# A dynamic and multi-responsive porous flexible metal–organic material

Mohana Shivanna [1], Qing-Yuan Yang[1], Alankriti Bajpai [1], Ewa Patyk-Kazmierczak[1] & Michael J. Zaworotko [1]

Stimuli responsive materials (SRMs) respond to environmental changes through chemical and/or structural transformations that can be triggered by interactions at solid-gas or solid-liquid interfaces, light, pressure or temperature. SRMs span compositions as diverse as organic polymers and porous inorganic solids such as zeolites. Metal–organic materials (MOMs), sustained by metal nodes and organic linker ligands are of special interest as SRMs. SR-MOMs have thus far tended to exhibit only one type of transformation, e.g. breathing, in response to one stimulus, e.g. pressure change. We report $[Zn_2(4,4'$-biphenyldicarboxylate$)_2(4,4'$-bis(4-pyridyl)biphenyl$)]_n$, an SR-MOM, which exhibits six distinct phases and four types of structural transformation in response to various stimuli. The observed structural transformations, breathing, structural isomerism, shape memory effect, and change in the level of interpenetration, are previously known individually but have not yet been reported to exist collectively in the same compound. The multi-dynamic nature of this SR-MOM is mainly characterised by using in-situ techniques.

[1] Department of Chemical Sciences, Bernal Institute, University of Limerick, Limerick V94 T9PX, Ireland. Correspondence and requests for materials should be addressed to M.J.Z. (email: Michael.Zaworotko@ul.ie)

S timuli responsive materials (SRMs) are broad in scope in terms of chemical composition and are of topical interest because of their potential applications[1–4]. An emerging class of SRMs is exemplified by porous metal organic materials (MOMs)[5,6], or porous coordination polymers (PCPs)[7,8] or metal–organic frameworks (MOFs)[9,10], which are extraordinarily diverse in composition thanks to their modularity and amenability to design from first principles[11]. SR-MOMs, a type of third generation PCPs (MOFs)[12,13], tend to undergo large structural flexibility[14–16] unlike more traditional classes of porous solids such as zeolites, for which structural transformations are typically less pronounced[17]. SR-MOMs also respond to a range of stimuli that are relevant to potential applications in gas storage[18], gas separations[19], drug release[20], molecular sensors[21] and catalysis[22]. Porous SR-MOMs that exhibit structural flexibility when exposed to gases/guests were introduced by the groups of S. Kitagawa and G. Ferey in 1998[12] and 2002[23], respectively, and are exemplified by materials such as MIL-53[23], MIL-88[24] and DMOF-1[25]. The latter is a square paddlewheel sustained pillared-layered network[26,27] that is prototypal for a range of variants[28].

The phase transformations of flexible SR-MOMs[7,29–32] are typically triggered by interfacial interactions and tend not to afford changes in network connectivity (topology)[33]. This is presumably because cleavage/regeneration of coordination bonds[33,34] and/or rearrangement of nodes would be required. SR-MOMs can, however, exhibit structural transformations, such as breathing[23], structural isomerism[34], shape memory effect[35,36] and

change in level of interpenetration[37] as a single response to a single stimulus.

## Results

**Synthesis and characterisation.** Herein, we report a new extended ligand (X-ligand) variant of DMOF-1, [Zn₂(4,4'-biphenyldicarboxylate)₂(4,4'-bis(4-pyridyl)biphenyl)]ₙ or X-pcu-1-Zn, was prepared by solvothermal reaction of X-ligands **L1**, (4,4'-bis(4-pyridyl)biphenyl) and **L2**, (biphenylene dicarboxylic acid) with $Zn(NO_3)_2 \cdot 6H_2O$ in N,N'-dimethylformamide (DMF) at 120 °C. Single crystals of the as-synthesised phase of X-pcu-1-Zn are designated X-pcu-1-Zn-3i-α. **L1** was introduced by Biradha et al.[38]. and prepared through Suzuki-Miyaura coupling, whereas **L2** is commercially available and was used as received from the vendor. The structure of X-pcu-1-Zn-3i-α determined by single-crystal x-ray diffraction (SCXRD) (Supplementary Table 1) reveals that it exhibits the expected primitive cubic, pcu, topology, threefold interpenetration and 46% free volume (Fig. 1 and Supplementary Fig. 1). Given the SR behaviour of its parent, DMOF-1, we subjected X-pcu-1-Zn-3i-α to a systematic investigation of its response to various stimuli, a study which revealed five other phases and four distinct SR effects.

**Breathing induced by heat or vacuum.** When a powdered sample of X-pcu-1-Zn-3i-α was heated under nitrogen, variable temperature powder x-ray diffraction (PXRD) measurements

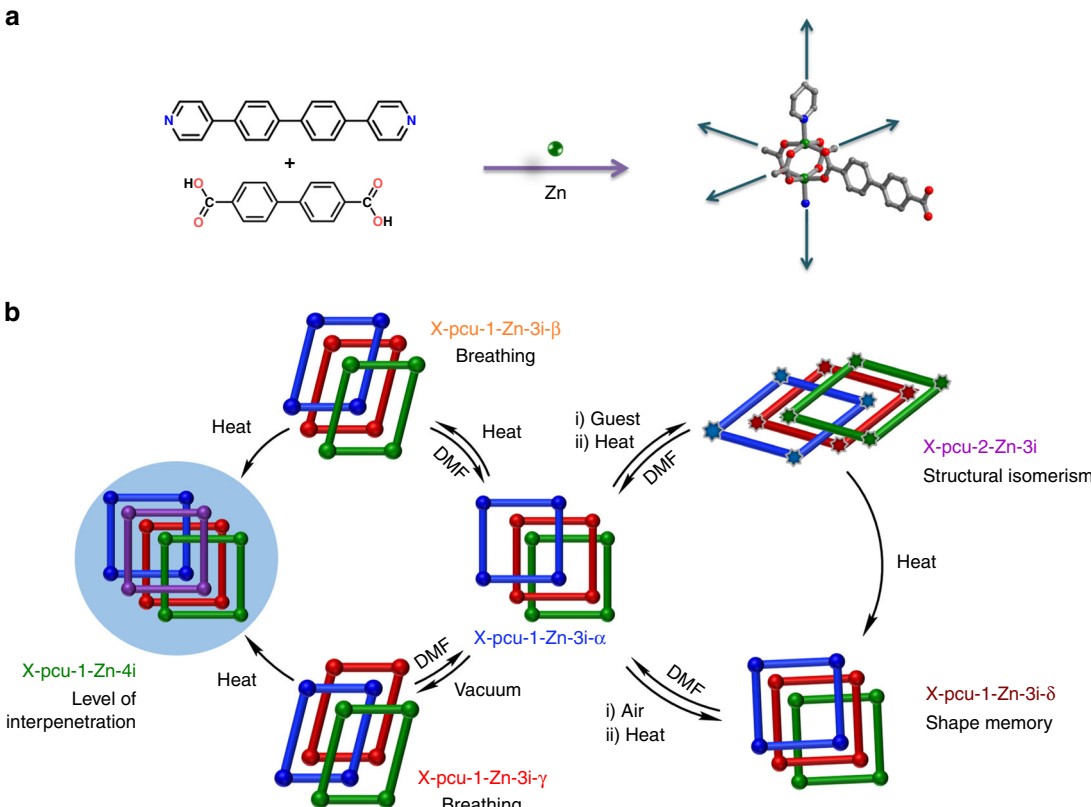

**Fig. 1** The multi-dynamic nature of X-pcu-1-Zn is exemplified by six structurally characterised phases. **a** Structures of 4,4'-bis(4-pyridyl)biphenyl **(L2)** and 4,4'-biphenyldicarboxylic acid **(L1)**, the pillaring of the 2D square-grid of [Zn₂(L1)₂] by **L2** affords the observed pcu topology network. **b** Breathing, X-pcu-1-Zn-3i-α transforms to X-pcu-1-Zn-3i-β and X-pcu-1-Zn-3i-γ; change in the level of interpenetration, X-pcu-1-Zn-3i-β and X-pcu-1-Zn-3i-γ convert to X-pcu-1-Zn-4i when heated at 130 °C; shape memory, X-pcu-1-Zn-3i-α exposed to air followed by heating affords X-pcu-1-Zn-3i-δ; structural isomerism: square-pyramidal Zn atoms in X-pcu-1-Zn-3i-α convert to distorted tetrahedral Zn atoms in X-pcu-2-Zn-3i after solvent exchange with acetonitrile followed by heating. X-pcu-2-Zn-3i, transforms into X-pcu-1-Zn-3i-δ upon further heating. With the exception of X-pcu-1-Zn-4i, all phases revert to X-pcu-1-Zn-3i-α after soaking in DMF for 5 min. X-pcu-1-Zn-4i, the densest phase, is the only phase that is thermodynamically stable under the conditions studied and is highlighted above. The interpenetrated networks are coloured green, maroon red, blue, and purple

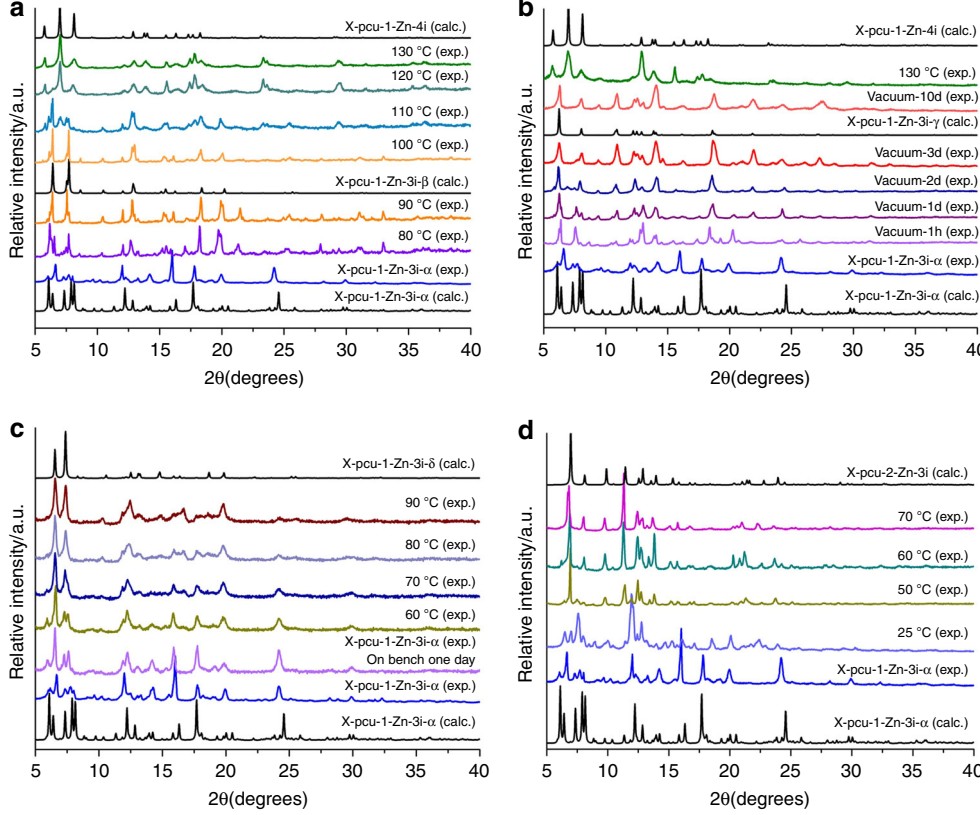

**Fig. 2** In situ variable temperature PXRD patterns reveal the following about X-pcu-1-Zn-3i-α. **a** Conversion to X-pcu-1-Zn-4i through X-pcu-1-Zn-3i-β. **b** Transformation into X-pcu-1-Zn-3i-γ under dynamic vacuum for 3 days and then to X-pcu-1-Zn-4i upon heating at 130 °C. **c** Formation of X-pcu-1-Zn-3i-δ when exposed to air for one day and upon heating. **d** Conversion to X-pcu-2-Zn-3i after solvent exchange with MeCN followed by heating at 70 °C. In addition, X-pcu-1-Zn-3i-α partially loses solvent at room temperature and forms a phase, X-pcu-1-Zn-3i-α′, with similar cell parameters as confirmed by SCXRD (SI). The experimental PXRD patterns of X-pcu-1-Zn-3i-α are therefore different in terms of intensity from the calculated PXRD patterns

indicated that a phase transformation had occurred at 90 °C (Fig. 2a). Single crystals of X-pcu-1-Zn-3i-α heated at 50 °C revealed a single-crystal-to-single-crystal, SCSC, transformation to X-pcu-1-Zn-3i-β (Supplementary Fig. 2), a phase in which free volume was reduced by 13% relative to X-pcu-1-Zn-3i-α.

When exposed to dynamic vacuum for three days, X-pcu-1-Zn-3i-α was found to transform to X-pcu-1-Zn-3i-γ (Fig. 2b), a phase in which free volume was reduced by 16% relative to X-pcu-1-Zn-3i-α. Both the β- and γ-phases reverted to the α-phase after soaking in DMF and retained macroscopic crystallinity through multiple transformation cycles (Supplementary Figs. 3, 4). X-pcu-1-Zn-3i-β and X-pcu-1-Zn-3i-γ undergo a breathing phenomenon, the typically observed transformation in SR-MOMs. Breathing can arise through structural distortions enabled by changes in bond angles and bond lengths[29] and has potential utility as it can enhance working capacity for gas storage[18].

**Structural isomerism induced by guest exchange.** Whereas distortion of networks through breathing/swelling has precedence in this class of MOMs[29], structural transformations that involve metal centres retaining their composition but changing their geometry, structural isomerism, are rare[33,34]. Soaking of X-pcu-1-Zn-3i-α in $CH_3CN$ and subsequent heating (70 °C, 10 min) resulted in transformation to X-pcu-2-Zn-3i, a process during which $Zn_2$(carboxylate)$_4$ square paddlewheel nodes (square-pyramidal Zn) transform to $Zn_2$(carboxylate)$_4$ nodes with tetrahedral Zn atoms (Supplementary Fig. 5). Transformation from X-pcu-1-Zn-3i-α to X-pcu-2-Zn-3i was monitored by in situ variable

temperature PXRD (Fig. 2d). X-pcu-2-Zn-3i reverts to X-pcu-1-Zn-3i-α when soaked in DMF for 5 min (Supplementary Fig. 6). The crystallinity of X-pcu-2-Zn-3i was maintained over two transformation cycles as confirmed by PXRD (Supplementary Fig. 7).

**Shape memory effect induced by exposure to humidity and heat.** Interpenetration occurs if two or more independent networks intertwine and cannot be separated without bond breakage[39]. However, external stimuli can induce networks to slip with respect to each other and change pore size and pore volume[40]. Exposure of X-pcu-1-Zn-3i-α to air for one day followed by heating to 90 °C under $N_2$ resulted in a transformation to new phase, X-pcu-1-Zn-3i-δ, as verified in situ variable temperature PXRD (Fig. 2c). The structure of X-pcu-1-Zn-3i-δ was determined by SCXRD (Supplementary Table 1), and it exhibits 37% guest accessible volume. X-pcu-1-Zn-3i-δ does not undergo further transformation upon exposure to heat or vacuum (Supplementary Fig. 8) although it reverts to X-pcu-1-Zn-3i-α when soaked in DMF (PXRD, Supplementary Fig. 9). X-pcu-1-Zn-3i-δ can also be prepared by heating X-pcu-2-Zn-3i at 90 °C for 12 h, as verified by PXRD (Supplementary Fig. 10). $N_2$ and $CO_2$ sorption isotherms for X-pcu-1-Zn-3i-δ prepared by the former method were collected at 77 and 195 K, respectively, and reveal a BET surface area (77 K $N_2$) of 556 m$^2$ g$^{-1}$ (Supplementary Fig. 11). PXRD of the bulk powder collected after gas sorption confirms that the phase remains as X-pcu-1-Zn-3i-δ (Supplementary Fig. 12). These results indicate that X-pcu-1-Zn-3i-δ exhibits a shape memory effect. To our knowledge, the only other

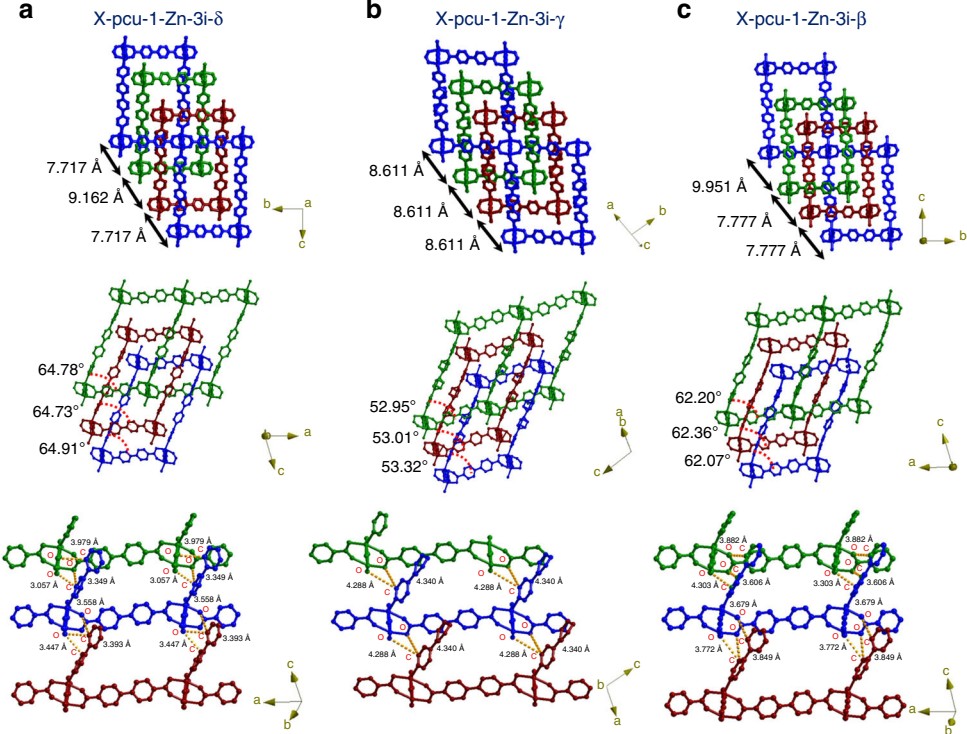

**Fig. 3** Structural insight into the shape memory phenomenon seen in X-pcu-1-Zn-3i-δ. The shape memory phase exhibits less strain (above and middle), shorter network-network contacts (below) and average repeat distances (above) than the other two phases. Threefold interpenetrated networks are coloured green, maroon, and blue. The bond angles and bond lengths are represented by red dashed (middle row) and yellow dashed lines (bottom row)

two example of a MOM that exhibits a shape memory effect, the related compound $[Cu_2(bdc)_2(bpy)]_n$, was reported by Sakata, Y. et al[35].and X-pcu-3-Zn by Shivanna, M. et al[36]. Whereas shape-memory effects are well-recognised in metal alloys, polymers, and ceramics[41,42]; they remain rare and poorly understood in crystalline porous materials.

To gain insight into the kinetically stable shape memory effect observed herein, the crystal structures of X-pcu-1-Zn-3i-δ, X-pcu-1-Zn-3i-γ and X-pcu-1-Zn-3i-β were analysed. In X-pcu-1-Zn-3i-δ, both the networks and the square paddlewheel nodes are less distorted ($\angle_{\text{linker-MBB-linker}}$ = 64.73–64.91°), (Fig. 3a middle, Supplementary Fig. 13) than in X-pcu-1-Zn-3i-γ and X-pcu-1-Zn-3i-β. In addition, there are short C–H⋯O ($D_{C⋯O}$ = 3.057 Å, 3.349 Å) contacts between the interpenetrated networks (Fig. 3a bottom). In X-pcu-1-Zn-3i-γ and X-pcu-1-Zn-3i-β the paddlewheel nodes exhibit $\angle_{\text{linker-MBB-linker}}$ angles of 52.95–53.32° and 62.07–62.36°, respectively (Fig. 3b middle, Supplementary Fig. 13), and there are longer contacts between adjacent networks ($D_{C⋯O}$ = 4.288 Å, 4.340 Å, Fig. 3b, bottom; $D_{C⋯O}$ = 3.303 Å, 3.849 Å, Fig. 3c middle and bottom, Supplementary Fig. 13). In three phases the measured repeating distances from square paddlewheel node to another square paddlewheel node by considering centroid at centre of node (Fig. 3 top row, Supplementary Fig. 14). Observed in changes in bond angles of coordination unit (Fig. 3 middle row, Supplementary Fig. 15). These structural data suggest that kinetically stable shape memory phases can be enabled by network-network interactions and reduced strain in the nodes and linkers.

**Change in level of interpenetration induced by heat and/or vacuum.** Another understudied type of transformation in SR-MOMs is change in the level of interpenetration[37,43]. In situ variable temperature SCXRD studies revealed that upon heating at 50 °C X-pcu-1-Zn-3i-α undergoes a SCSC phase

transformation to X-pcu-1-Zn-3i-β. Increasing the temperature to 130 °C results in transformation to X-pcu-1-Zn-4i, a fourfold interpenetrated phase (Supplementary Fig. 2). Crystallinity is reduced but the domains remain large enough to enable structure determination. The square paddlewheel node in X-pcu-1-Zn-4i is retained but the space group changed from P-1 to C2/c (Supplementary Table 1), and the guest accessible volume reduced from 46 to 28%. X-pcu-1-Zn-3i-γ also transforms to X-pcu-1-Zn-4i at 130 °C (Fig. 2b). The transformation to X-pcu-1-Zn-4i appears to be irreversible; heat, exposure to solvent and exposure to vacuum did not induce further phase changes (Supplementary Figs. 16, 20). The transformation of X-pcu-1-Zn-3i-α to X-pcu-1-Zn-4i was also monitored by optical thermal microscopy on a single crystal that had been face-indexed. The largest face underwent a 12% reduction in size upon desolvation at 100 °C for 30 min. Face-indexing revealed that single-crystals crack along the (101) face, suggesting that this plane is affected by transformation to X-pcu-1-Zn-4i. The (101) plane has the highest density of metal coordination centres (Fig. 4h and Supplementary Fig. 19). $N_2$ and $CO_2$ sorption isotherms were collected upon X-pcu-1-Zn-4i at 77 and 195 K, respectively. The BET surface area calculated from the 77 K $N_2$ isotherm is 432 $m^2 g^{-1}$ (Supplementary Fig. 17).

A mechanism that has been proposed for interpenetration changes in MOMs suggests that biphenylene carboxylate-linked moieties could bend laterally while maintaining their connectivity during the conversion[37]. Should that mechanism apply to X-pcu-1-Zn-3i-α then heating at 50 °C or exposure to dynamic vacuum would lead to loss of DMF (Supplementary Fig. 18c, d) and enable paddlewheel moieties to be in closer proximity. Further heating and solvent loss brings independent networks even closer (Fig. 4a), which facilitates increase in degree of interpenetration from threefold to fourfold through breakage/regeneration of coordination bonds.

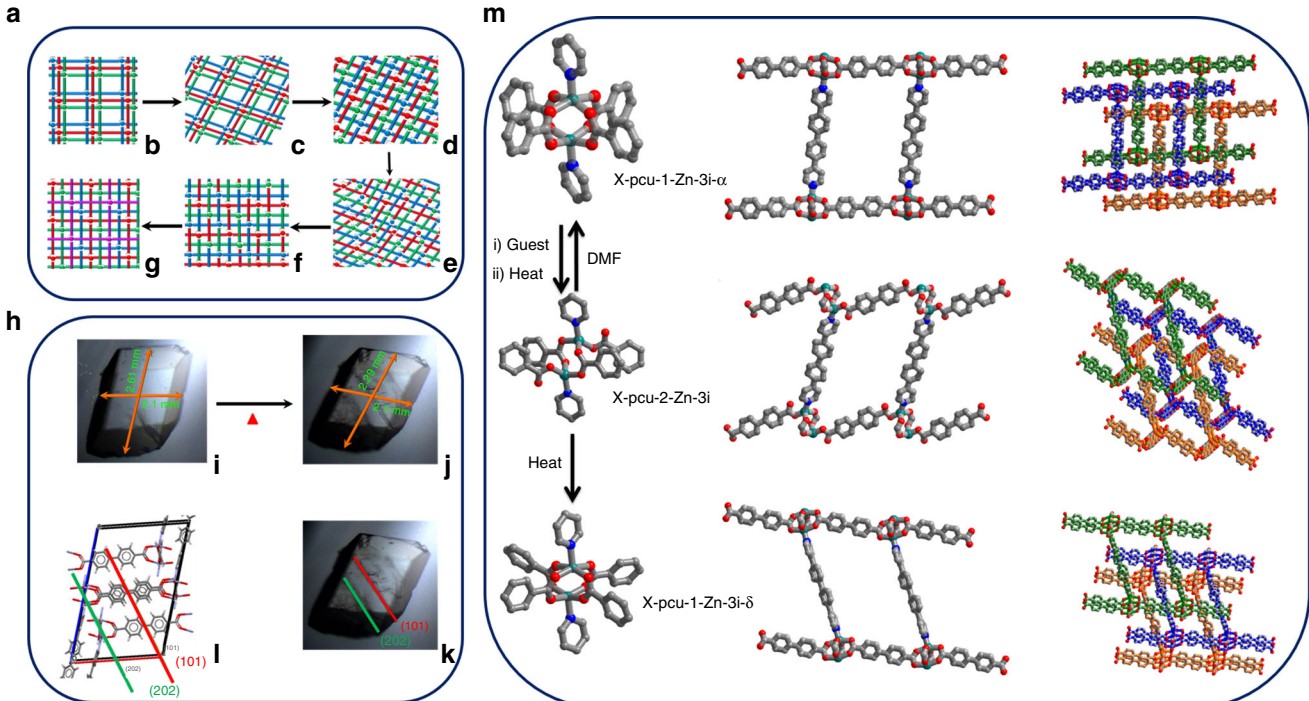

**Fig. 4** Structural transformation. **a** Proposed mechanism[37] for conversion of X-pcu-1-Zn-3i-α to X-pcu-1-Zn-4i: **b** 3-fold interpenetrated pcu nets are offset; **c** structure in between α- and β-phase with networks tilted but still offset; **d** centered interpenetration of networks; **e** distortion leads to breakage of metal carboxylate and metal nitrogen coordination bonds; **f** non-coordinated carboxylate or nitrogen moieties bind to available metal centres; **g** 4-fold interpenetrated network. The interpenetrated networks are coloured green, red, blue and purple. **h** Thermal microscopy images: **i** as-synthesised crystal of X-pcu-1-Zn-3i-α; **j** X-pcu-1-Zn-3i-α after desolvation at 100 °C for 30 min; **k** and **l** cracks along (101) and parallel to (101) that occur as X-pcu-1-Zn-3i-α transforms to X-pcu-1-Zn-4i. **m** isomerisation of the node from square paddlewheel geometry (X-pcu-1-Zn-3i-α) to distorted tetrahedral geometry (X-pcu-2-Zn-3i) and then back to square paddlewheel (X-pcu-1-Zn-3i-δ). Three-fold Interpenetrated networks are presented in green, blue and yellow. Colour scheme: C, grey; O, oxygen; N, blue; Zn, dark green.

In summary, we report herein a porous SR-MOM, X-pcu-1-Zn, that exhibits multi-dynamic behaviour in that it undergoes four types of SR transformation: breathing, structural isomerism, shape memory effect, and change in level of interpenetration. The structural insight gained from the characterisation of six distinct phases enables an understanding of what drives the shape memory and interpenetration change effects, both of which are understudied and poorly understood phenomena. Notably, the stimuli used by us are representative of methods commonly employed to activate porous materials. Should the type of behaviour observed herein be common, it is likely to explain discrepancies between single crystal structures and PXRD patterns of activated porous materials. Our observations highlight the need for researchers to use in situ characterisation tools when evaluating the properties of porous MOMs since bulk property measurements after activation might often be conducted upon a MOM that is quite different in structure from that of the as-synthesised MOM.

## Methods

**Synthesis of [Zn₂(4,4′-biphenyldicarboxylate)₂(4,4′-bis(4-pyridyl)biphenyl)]ₙ ·10DMF (X-pcu-1-Zn-3i-α).** Commercially available reagents were purchased in high purity grade and used as received. A mixture of $Zn(NO_3)_2 \cdot 6H_2O$ (14.8 mg, 0.05 mmol; purchased from Sigma-Aldrich), 4,4′-biphenyldicarboxylate, **L2** (12.11 mg, 0.05 mmol; purchased from Sigma-Aldrich), 4,4′-bis(4-pyridyl) biphenyl, **L1**(7.7 mg, 0.025 mmol) in DMF (3 mL) or DEF/MeOH (2:1 v/v) was added to a 14 mL glass vial. The vial was capped tightly and placed in an oven at 120 °C for 24 h to yield block-shaped colourless crystals. The contents of the vial were cooled to room temperature. Crystals were filtered and washed with 5 mL of DMF or DEF three times. Yield (22 mg, 64%) based upon the limiting reagent being **L1**.

**Synthesis of 4,4′-bis(4-pyridyl) biphenyl (L1).** The 4,4′-bis(4-pyridyl)biphenyl was synthesised by twofold Pd⁰-catalysed Suzuki-Miyaura coupling of 4-pyridynylboronic acid with 4,4′-dibromobiphenyl using a modified procedure[44]. A 250 mL oven-dried two-necked round bottom flask was cooled under N₂ atmosphere and charged with 4,4′-dibromobiphenyl (2.0 g, 6.41 mmol), 4-pyridinylboronic acid (2.34 g, 19.11 mmol), Pd(PPh₃)₄ (0.37 g, 0.32 mmol), powdered NaOH (1.03 g, 25.64 mmol), 50 mL of toluene, 30 mL of EtOH, and 15 mL of distilled water. The resultant reaction mixture was refluxed under a positive pressure of N₂ at 110 °C for 2 days. The change in the colour of reaction mixture from yellow to dark brown indicated completion of the reaction that was further verified by thin layer chromatography analysis. After this, the reaction mixture was cooled, contents concentrated in vacuo and the residue extracted using CHCl₃/brine solution. The organic phase was dried over anhyd Na₂SO₄ and concentrated in vacuo. Recrystallisation using EtOAc afforded a white solid in 95% yield (1.92 g, 6.03 mmol). ¹H NMR (CDCl₃, 270 MHz) δ 8.68 (d, J = 5.9 Hz, 4H), 7.74–7.76 (m, 8H), 7.55 (d, J = 5.9 Hz, 4H).

**Preparation of [Zn₂(L2)₂(L1)]·6DMF (X-pcu-1-Zn-3i-β).** As synthesised crystals of X-pcu-1-Zn-3i-α were heated at 90 °C for 10 min under N₂ atmosphere or in the oven.

**Preparation of [Zn₂(L2)₂(L1)]·DMF (X-pcu-1-Zn-3i-γ).** As synthesised crystals of X-pcu-1-Zn-3i-α were evacuated under dynamic vacuum 3d, rt.

**Preparation of [Zn₂(L2)₂(L1)]·3H₂O, 2DMF (X-pcu-1-Zn-3i-δ).** As synthesised crystals of X-pcu-1-Zn-3i-α were exposed to air on bench one or two days then heated at 90 °C for 10 min under N₂ atmosphere.

**Preparation of [Zn₂(L2)₂(L1)] (X-pcu-2-Zn-3i).** The pure phase of X-pcu-2-Zn-3i was prepared by solvent exchange of X-pcu-1-Zn-3i-α with acetonitrile three times a day for two days. Solvent exchanged crystals were then heated at 70 °C for 10 min under N₂ atmosphere.

**Preparation of [Zn$_2$(L2)$_2$(L1)] (X-pcu-1-Zn-4i)**. Crystals of the X-pcu-1-Zn-4i were prepared by heating X-pcu-1-Zn-3i-α under dynamic vacuum at 130 °C for 12 h or heating X-pcu-1-Zn-3i-α in the oven at 130 °C for 12 h.

**In-situ variable temperature PXRD experiments**. PXRD experiments were carried out on a PANalytical Empyrean™ diffractometer equipped with a PIXcel3D detector and Cu K-alpha radiation ($\lambda_\alpha = 1.5418$ Å). The diffractometer was operated at 40 kV and 40 mA and experiments were conducted in continuous scanning mode with the goniometer in the theta-theta orientation. The as synthesised crystals were ground into a fine powder, placed on zero background sample holder, mounted to a bracket flat sample stage and exposed to x-ray radiation. In a typical experiment, data was collected via a continuous scan in the range of 4°–45° (2θ) with a step-size of 0.02° at a scan rate of 0.1° min$^{-1}$. In situ variable temperature PXRD was carried out under N$_2$ atmosphere with increasing temperature of 10 °C for every 10 min interval of time. An external water-cooling system had been adopted to maintain the temperature. X'Pert HighScore Plus™ software V 4.1 (PANalytical, The Netherlands) was used for evaluating and analysing raw data.

**SCXRD experiments**. Single-crystal diffraction data were collected on a Bruker Quest diffractometer equipped with a CMOS detector and IμS microfocus X-ray source (Cu $K_\alpha$, $\lambda = 1.5418$ Å). Absorption corrections were applied by using the multi-scan program SADABS. Indexing was performed using APEX2 [Bruker, 2012][45] (Difference Vectors method). Data integration and reduction were performed using SaintPlus [Bruker, 2012]. Space group was determined using XPREP implemented in APEX2 [Bruker, 2012]. Structural solution and refinement against $F$ were carried out using the SHELXL programs.

**In-situ SC–SC XRD experiments**. In situ SC–SC data were collected by choosing a single crystal, glueing it to a glass fibre and inserting it into a glass capillary. The capillary dimensions were a wall thickness of 0.01 mm and an outer diameter of 0.2 mm. The instrument was thermostatically set at 298 K and a single crystal in a capillary was mounted on the goniometer head. X-ray diffraction data were collected and yielded the same cell parameters as the as-synthesised form of X-pcu-1-Zn-3i-α. Temperature was then set to 298, 323, 363, and 403 K for approximately 4 h in order to determine the unit cell parameters at each temperature. Full sets of single crystal data were collected at three temperature (298, 323, and 403 K) and the structures were refined.

**Thermal microscopy study**. A Zeiss Axio Imager externally connected to a Peltier controller (Linkam T95) and heating stage was used. The system has a temperature range of −20 to 120 °C. The rate of heating/cooling can be varied within the range 0.1 °C per minute to 20 °C per minute. A single crystal of X-pcu-1-Zn-3i-α was placed on glass slide with a length scale. The sample was heated from 20 to 100 °C, with a heating rate of 1 °C per minute and held constant at 100 °C for 30 min. Diagonal cracks had by then formed and the temperature was then increased to 120 °C. Temperature was held at 120 °C for 1 h. The length along two axes of the single crystal at 100 and 120 °C were compared with those of the room temperature crystal. Face indexing of the single crystal after it was heated at 100 °C revealed that cracks occurred at the (101) face.

**Thermogravimetric analysis (TGA)**. TGA was carried out under nitrogen using a TA Q50 instrument. Experiments were conducted using platinum pans and with a flow rate of 60 mL min$^{-1}$ of nitrogen gas. The data were collected in the high resolution dynamic mode with a sensitivity of 1.0, a resolution of 4.0, and a temperature ramp of 10 °C min$^{-1}$ up to 550 °C. The data were analysed using the T.A. universal analysis suite for windows XP/ Vista version 4.5 A.

**Data availability**. Supplementary crystallographic data for this manuscript has been deposited at the Cambridge Crystallographic Data Centre under deposition numbers CCDC 1524722–1524727 and CCDC 1845763. These data can be obtained free of charge from http://www.ccdc.cam.ac.uk/data_request/cif.

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

## Acknowledgements

The authors thank the Science Foundation Ireland for funding of this research (SFI Award 13/RP/B2549). We thank to Drs. John J. Perry IV and David G. Madden for discussions and inputs at early stages of drafting. We thank Dr. Matteo Lusi for assistance with in situ single crystal measurements.

## Author contributions

M.J.Z. and M.S. conceived the experiments and designed the study. M.S. carried out the material syntheses and measurements. M.S., Q.Y.Y. participated characterisation of the phases reported herein (SCXRD, PXRD and gas sorption). A.B. developed the synthetic method to prepare L1. E.P.K refined the crystal structures. M.J.Z., Q.Y.Y., and M.S. wrote the manuscript. All authors discussed the results and contributed to the interpretation of data. All authors contributed to editing the manuscript.

## Additional information

**Competing interests:** The authors declare no competing interests.

