## [Peer Review File · Nature Communications]

Reviewers' comments:

Reviewer #1 (Remarks to the Author):

This manuscript reported an interesting phenomenon regarding the multiple structural transformations of a flexible interpenetrated metal-organic framework under multiple types of stimuli. This unique system involves six distinct phases (I see six from Figure 1 but the authors mentioned five in the abstract !) that can be converted to each other through four types of structural transformation, namely breathing, structural isomerism, shape memory effect and change in the level of interpenetration. As the authors stated, although all the above-mentioned transformations have been reported individually before, some more frequently (such as breathing) and others relatively rare (such as the shape memory effect recently reported by the same group), there is no example of such that can integrate all four transformations into one system so far. The structural transformations were well characterized by the in-situ variable temperature PXRD and in-situ SC-SC XRD measurements. I think it is a great piece of work, which can offer new insights about the structural transformation process in a flexible MOF and interpenetrated MOF. I recommend its publication in Nature Communications if the following technical issues can be appropriately addressed.

- 1) As I mentioned earlier, is there five or six distinct phases? Please confirm.
- 2) Figure 1, it would be nice if the condition for the reverse transformation to X-pcu-1-Zn-3i-a is also shown.
- 3) For the powder XRD of sample x-pcu-1-Zn-3i-a, there is unignorable mismatch between the experimental and simulated patterns (for example, the second major peak around 6.5-7 region and many peaks after 12 θ). How was the PXRD done for this sample? It seems there might be some structural change on this sample already.
- 4) Figure 3, the first row. Given it is a 3-interpenetrated MOF, is it true that three distances between four adjacent nets should be given to depict the full feature of a repeating unit?
- 5) For Figure 4a, it seems the authors followed the style of Figure 2 in Ref. 36 to explain the transformation mechanism. However, I feel it's not clear enough, even I tried hard reading it. Especially, a5 looks no difference to a1, a2 or a3, except the relative distance and tilting angle. And later it suddenly turns into a6. The bond breaking and reformation can't be seen.
- 6) The thermal microscopy (Figure 2b) is nice, but I feel it's better to label the scale (such as 2.6x2.1 mm) directly on the crystal.

Reviewer #2 (Remarks to the Author):

The article 'A multi-dynamic and multi-responsive porous flexible metal-organic material' by Michael J. Zaworotko and researchers reports a fascinating stimuli-responsive Metal-organic material (SR MOM) that can show collectively many functions, such as (thermally induced) structural transformations, breathing, structural isomerism, shape memory effect and change in the level of interpenetration. These researchers have put in a lot of work in studying and characterizing this system which changes with various stimuli.

The strong contending point seems to be that this SR MOM is 'all-in-one' system, whereas systems reported earlier are known to perform only one of the functions. Of course, this is a very impressive structural assembly, but the work has not yet demonstrated any significant application that would have a wider impact. Also, in my opinion, (a) single crystal structure refinements (and disorder modeling) need a better finish and (b) graphics presented in the text needs revision for clearer understanding.

If authors are willing to give some thought to (a) and (b) above, this paper may well become suitable for publication – but, as it stands now, in a specialized journal, such as nature chemistry.

Single crystal structure refinements:

In the structure of X-pcu-1-Zn-3i- α , the solvent molecules show unusually large thermal ellipsoids (Fig. 1 from the deposited cif). Is it necessary to assign them full occupancies? Why were they not treated as partially occupied / disordered? In fact, wouldn't this feature be rationalized better with the solvent exchange (with acetonitrile) and conversion to X-pcu-2-Zn-3i?

Thermal ellipsoids in both the ligands L1 and L2 show inadequate modeling of the disorder /or application of restraints/constraints in the refinement. For example, in structures of X-pcu-1-Zn-3i- α (Fig. 2, all figures are generated from the deposited cifs), X-pcu-1-Zn-3i- β (Fig. 3) and X-pcu-1-Zn-3i- δ (Fig. 5) the elongated ellipsoids at one edge for the ligand L1 clearly indicate the rotation(s) of the aromatic ring about the molecular axis. Also in the structure of X-pcu-1-Zn-3i- β (Fig. 4), the ligand L2 exhibits ellipsoids that require some attention. It is not a concern about the crystallographic refinement alone, but it is about interpretations that could be derived on dynamical modes, conformational flexibilities of the bound ligands in the network.

Lastly, the structure refinement of X-pcu-1-Zn-4i has converged to a very high R factor (18.76, $R_w = 42.59$). What has contributed to such high R values?

Figures presented in the text:

A minor comment – (a) and (b) may be interchanged in Fig. 1? So, the components forming the complex are shown in (a) and then the conversion cycles are shown in the following figure (b)

Fig. 1(a) The figure shows the conversion cycles clearly, but a reader may be left wondering looking at the little figures of the networks (in different colours) what are the structural differences showed that are associated with each of these phases! Could anything be added in the figure caption and/or alternative representation of grids may be used?

Fig. 3 It is not clear from the figure (or even from the text) what points are used to measure the distances and angles marked on the figure. What are the atoms used on L1a or L2a in bond angle (for example, \angle L1a-C-L2a in Fig. S2(a))? Similarly, it is not clear what atoms or points are used to measure the distances. These may be made clear (could not find it easily in the text or in the supplementary material).

Fig. 4 The figure attempts to show the structural transformations, but representations of grids in part (a) are not very clear. Authors may think of providing alternative representations (perhaps showing the metal centers with small spheres in these and L1, L2 connectivity in different colours?)

In summary, authors may look at their draft again in the light of the comments made above. The intention is to make this article a finely finished product where a non-specialist scientist can grasp the structural dynamics in the crystal that takes place with various stimuli.

Reviewer #3 (Remarks to the Author):

The work by Zaworotko describes a new MOF with interesting structural changes observed upon various stimuli. Zaworotko is certainly a leading figure in the structural analysis of PCPs. The most interesting point of this contribution is the proposed mechanism for changes in interpenetration based on cluster restructuring. The materials have a relatively low porosity of about 500 m²g⁻¹. The strength of this study is the in depth crystallographic analysis, in particular the single crystal studies are deep. The refinement of the PXRDs could be improved (see comments). The selectivity for a number of gases is given, however, relatively few gases show MOF flexibility in the presence of gases. In this context, probably the reference to 3rd generation MOFs is

inadequate, as the host structural changes are not a specific guest response.

However, the crystallographic study gives valuable insights into the diverse transformation mechanisms. In particular the deformation of paddle wheel units is seen as a motif allowing for interpenetration changes.

Overall, a valuable work for the MOF community and in crystal engineering. Congratulations!

Minor points:

- Title: The repeated use of „multi“ gives an overselling impression
- How well is the assignment of „degree of interpenetration“ from powder data? In principal also partial interpenetration may be possible, making the picture more complex (not “black and white” so to say).
- Fig S7 reveals also significant differences in peak positions for calc. vs. Exp. Is this I.c. shift taken into account?

- SR-MOMs: Maybe here the established term „3rd generation PCPs (MOFs)“ would be preferred.
- Fig.1: It would be useful to get an understanding which phases are considered as metastable, and which ones as „stable“ from a thermodynamic perspective.
- „Exposure of X-pcu-1-Zn-3i-a to air“: what is the role of air exposure? Is moisture and water coordination a trigger?
- What is the role of crystallite size in these phenomena?
- S14: mmHg \diamond Pa (SI)

Response to reviewer comments

We are grateful to the three referees for their time and constructive comments. We have addressed these comments as detailed below and believe that the revised manuscript is improved thanks to their comments. For your convenience, we have highlighted all revisions in the revised manuscript and SI.

Reviewer: 1

This manuscript reported an interesting phenomenon regarding the multiple structural transformations of a flexible interpenetrated metal-organic framework under multiple types of stimuli. This unique system involves six distinct phases (I see six from Figure 1 but the authors mentioned five in the abstract !) that can be converted to each other through four types of structural transformation, namely breathing, structural isomerism, shape memory effect and change in the level of interpenetration. As the authors stated, although all the above-mentioned transformations have been reported individually before, some more frequently (such as breathing) and others relatively rare (such as the shape memory effect recently reported by the same group), there is no example of such that can integrate all four transformations into one system so far. The structural transformations were well characterized by the in-situ variable temperature PXRD and in-situ SC-SC XRD measurements.

I think it is a great piece of work, which can offer new insights about the structural transformation process in a flexible MOF and interpenetrated MOF. I recommend its publication in Nature Communications if the following technical issues can be appropriately addressed.

- 1) As I mentioned earlier, is there five or six distinct phases? Please confirm.

Response. The reviewer is correct. There are indeed six distinct phases and the text has been revised accordingly.

- 2) Figure 1, it would be nice if the condition for the reverse transformation to X-pcu-1-Zn-3i- α is also shown.

Response. We agree. Figure 1 has been revised.

- 3) For the powder XRD of sample X-pcu-1-Zn-3i- α , there is unignorable mismatch between the experimental and simulated patterns (for example, the second major peak around 6.5-7 region and many peaks after 12 θ). How was the PXRD done for this sample? It seems there might be some structural change on this sample already.

Response. The referee is correct. We attribute the PXRD differences to partial loss of solvent since unit cell parameters of single crystals exposed to air are unchanged. The following has been added to the caption of Figure 2 and the unit cell parameters of the partially desolvated phase were determined and are presented in Table S1: *In addition, X-pcu-1-Zn-3i- α partially loses solvent at room temperature and forms a phase, X-pcu-1-Zn-3i- α' , with similar cell parameters as confirmed by SCXRD (SI). The experimental PXRD patterns of X-pcu-1-Zn-3i- α are therefore different in terms of intensity from the calculated PXRD patterns.*

- 4) Figure 3, the first row. Given it is a 3-interpenetrated MOF, is it true that three distances between four adjacent nets should be given to depict the full feature of a repeating unit?

Response. In order to avoid confusion, we have modified Figures 3 and S2c to address this matter. The nature of the interpenetration is such that crystallographic symmetry requires two repeat distances for the delta and beta phases and only one for the gamma phase. Figure 3 (caption and graphic) and Figure S2c (graphic) have been modified.

- 5) For Figure 4a, it seems the authors followed the style of Figure 2 in Ref. 36 to explain the transformation mechanism. However, I feel it's not clear enough, even I tried hard reading it. Especially, a5 looks no difference to a1, a2 or a3, except the relative distance and tilting angle. And later it suddenly turns into a6. The bond breaking and reformation can't be seen.

Response. We agree. Figure 4a has been modified following the reviewer's suggestions.

- 6) The thermal microscopy (Figure 2b) is nice, but I feel it's better to label the scale (such as 2.6x2.1 mm) directly on the crystal.

Response. We agree. Figure 2b has been modified.

Reviewer: 2

The article 'A multi-dynamic and multi-responsive porous flexible metal-organic material' by Michael J. Zaworotko and researchers reports a fascinating stimuli-responsive Metal-organic material (SR MOM) that can show collectively many functions, such as (thermally induced) structural transformations, breathing, structural isomerism, shape memory effect and change in the level of interpenetration. These researchers have put in a lot of work in studying and characterizing this system which changes with various stimuli. The strong contending point seems to be that this SR MOM is 'all-in-one' system, whereas systems reported earlier are known to perform only one of the functions. Of course, this is a very impressive structural assembly, but the work has not yet demonstrated any significant application that would have a wider impact. Also, in my opinion, (a) single crystal structure refinements (and disorder modeling) need a better finish and (b) graphics presented in the text needs revision for clearer understanding.

If authors are willing to give some thought to (a) and (b) above, this paper may well become suitable for publication – but, as it stands now, in a specialized journal, such as nature chemistry.

- 1) Single crystal structure refinements: In the structure of X-pcu-1-Zn-3i- α , the solvent molecules show unusually large thermal ellipsoids (Fig. 1 from the deposited cif). Is it necessary to assign them full occupancies? Why were they not treated as partially occupied / disordered? In fact, wouldn't this feature be rationalized better with the solvent exchange (with acetonitrile) and conversion to X-pcu-2-Zn-3i?

Response. We attribute the large thermal ellipsoids to hard-to-model disorder of solvent molecules. Further, based on TGA data, there should be 10 molecules present in the asymmetric unit, but only 7 molecules could be located and refined. Residual electron density observed in close proximity to DMF molecules coupled with their large thermal ellipsoids suggests that each of the 7 molecules is disordered. We made several attempts to model the disorder but refinement resulted in unreasonable structures. It is therefore plausible that the disorder occurred over more than two positions around the assigned position of the DMF molecules. In addition, partial occupancies were assigned to each DMF molecule and no significant changes in thermal ellipsoids were observed. In

order to present a structure that is as chemically accurate as possible, full occupancy of DMF molecules was therefore assigned. The discrepancy in the number of molecules in asymmetric unit, as well as large thermal ellipsoids is addressed in the Supporting Information and CIF files.

- 2) Thermal ellipsoids in both the ligands L1 and L2 show inadequate modeling of the disorder /or application of restraints/constraints in the refinement. For example, in structures of X-pcu-1-Zn-3i- α (Fig. 2, all figures are generated from the deposited cifs), X-pcu-1-Zn-3i- β (Fig. 3) and X-pcu-1-Zn-3i- δ (Fig. 5) the elongated ellipsoids at one edge for the ligand L1 clearly indicate the rotation(s) of the aromatic ring about the molecular axis. Also in the structure of X-pcu-1-Zn-3i- β (Fig. 4), the ligand L2 exhibits ellipsoids that require some attention. It is not a concern about the crystallographic refinement alone, but it is about interpretations that could be derived on dynamical modes, conformational flexibilities of the bound ligands in the network.

Response. Disorder was modelled as suggested where appropriate and possible. However, the shape and size of thermal ellipsoids for the disorder models indicates that the rings in question are disordered over several positions between two extremes rather than simply between two positions. For the same reason, anisotropic refinement of the disorder of some of the rings was not possible (unreasonable ellipsoids, non-positive definite U for some atoms). In order to maintain the anisotropic refinement for the whole structure, without adding strong restraints and constraints, in some cases disorder was not modelled. CIFs for modified refinements have been submitted to the CCDC.

- 3) Lastly, the structure refinement of X-pcu-1-Zn-4i has converged to a very high R factor (18.76, $R_w = 42.59$). What has contributed to such high R values?

Response. X-pcu-1-Zn-4i was obtained after heating the sample. This process affected the crystal quality and the quality of collected data, and resulted in high values of the R factors.

Figures presented in the text:

- 4) A minor comment – (a) and (b) may be interchanged in Fig. 1? So, the components forming the complex are shown in (a) and then the conversion cycles are shown in the following figure (b).

Response. We agree. Figure 1 has been modified as requested.

- 5) Fig. 1(a) The figure shows the conversion cycles clearly, but a reader may be left wondering looking at the little figures of the networks (in different colours) what are the structural differences showed that are associated with each of these phases! Could anything be added in the figure caption and/or alternative representation of grids may be used?

Response. We agree. Figure 1 (now Figure 1b) has been modified as suggested by the reviewer.

- 6) Fig. 3 It is not clear from the figure (or even from the text) what points are used to measure the distances and angles marked on the figure. What are the atoms used on L1a or L2a in bond angle (for example, \angle L1a-C-L2a in Fig. S2(a))? Similarly, it is not clear what atoms or points are used to

measure the distances. These may be made clear (could not find it easily in the text or in the supplementary material).

Response. We agree. Figure 3 and Figures S2a, S2c and S2d have been modified.

7) Fig. 4 The figure attempts to show the structural transformations, but representations of grids in part (a) are not very clear. Authors may think of providing alternative representations (perhaps showing the metal centers with small spheres in these and L1, L2 connectivity in different colours?)

Response. We agree. Figure 4a has modified accordingly.

Reviewer: 3

The work by Zaworotko describes a new MOF with interesting structural changes observed upon various stimuli. Zaworotko is certainly a leading figure in the structural analysis of PCPs. The most interesting point of this contribution is the proposed mechanism for changes in interpenetration based on cluster restructuring. The materials have a relatively low porosity of about 500 m²g⁻¹. The strength of this study is the in depth crystallographic analysis, in particular the single crystal studies are deep. The refinement of the PXRDs could be improved (see comments). The selectivity for a number of gases is given, however, relatively few gases show MOF flexibility in the presence of gases. In this context, probably the reference to 3rd generation MOFs is inadequate, as the host structural changes are not a specific guest response. However, the crystallographic study gives valuable insights into the diverse transformation mechanisms. In particular the deformation of paddle wheel units is seen as a motif allowing for interpenetration changes.

Overall, a valuable work for the MOF community and in crystal engineering. Congratulations!

1) Title: The repeated use of “multi” gives an overselling impression

Response. We agree. The title has been changed to **“A dynamic and multi-responsive porous flexible metal–organic material”**

2) How well is the assignment of “degree of interpenetration“ from powder data? In principal also partial interpenetration may be possible, making the picture more complex (not “black and white” so to say).

Response. We rely on both PXRD and SCXRD to assign the degree of interpenetration. With the exception of the alpha phase (see response to reviewer 1, there is a good match between calculated and experimental PXRD patterns. Further, there are no anomalies in gas sorption profiles. We are therefore confident that partial interpenetration is not an issue herein.

3) Fig S7 reveals also significant differences in peak positions for calc. vs. Exp. Is this l.c. shift taken into account?

Response. We agree. The matter of the significant different with respect to the alpha phase was addressed in comments to reviewer 1. With respect to minor differences, single crystal data was

collected at 100K but matched with experimental pxrd collected at 298K, 343K and 363K respectively.

4) SR-MOMs: Maybe here the established term “3rd generation PCPs (MOFs)” would be preferred.

Response. We have inserted the following in to the text of the revised manuscript: SR-MOMs, a type of 3rd generation PCPs (MOFs),²¹.

5) Fig.1: It would be useful to get an understanding which phases are considered as metastable, and which ones as “stable” from a thermodynamic perspective.

Response. We agree. The 4-fold interpenetrated phase is the only phase that is stable under all of the conditions studied and we have now address this matter in the caption to Figure 1.

6) “Exposure of X-pcu-1-Zn-3i- α to air”: what is the role of air exposure? Is moisture and water coordination a trigger?

Response. Upon expose to air, there is gradual replacement of DMF by water. A similar effect was observed for MIL-53 when exposed to air. The relevant paper, *Angew. Chem. Int. Ed.* 2006, 45, 7751–7754, is cited in the Introduction.

7) What is the role of crystallite size in these phenomena?

Response. We synthesised several batches of each sample and with varying particle sizes. No difference in sorption properties was observed.

8) S14: mmHg or kPa (SI).

Response. Figure S14 has been modified accordingly.

REVIEWERS' COMMENTS:

Reviewer #1 (Remarks to the Author):

In the revised version, the authors have done a great job to address my questions as well as questions from other referees. I am very happy with the responses and corresponding changes to my question 1-4&6. For Figure 4a, it is still a bit difficult for me to read, although efforts for improvement can be clearly seen. Considering the challenging nature to express such structural transformations, I think this figure is acceptable now as is. Given the technical issues have been cleared, I think it is suitable for publication now.

Again, I think this is a great piece of work by Dr. Zaworotko and coworkers. While soft MOFs and interpenetrated MOFs have been frequently reported. There is no study so far on a system that can integrate so many types of structural transformations. These transformations have been well identified using solid proof using crystallography technique. This interesting example provides us great opportunity to better understand the MOF materials.

Reviewer #2 (Remarks to the Author):

Author's explanations for this reviewer's comments on single crystal refinements and changes made by them in the graphics are satisfactory. Having gone through the reviewer's comments and author's detailed responses and revisions made in the manuscript, this article in its present form has become suitable for publication - in my view -in nature chemistry.